# Identification of Potential Key lncRNAs in the Context of Mouse Myeloid Differentiation by Systematic Transcriptomics Analysis

**DOI:** 10.3390/genes12050630

**Published:** 2021-04-23

**Authors:** Yongqing Lan, Meng Li, Shuangli Mi

**Affiliations:** 1Key Laboratory of Genomic and Precision Medicine, Beijing Institute of Genomics, Chinese Academy of Sciences, Beijing 100101, China; lanyq@big.ac.cn (Y.L.); limeng@big.ac.cn (M.L.); 2China National Center for Bioinformation, Beijing 100101, China; 3University of Chinese Academy of Sciences, Beijing 100049, China

**Keywords:** myeloid differentiation, lncRNAs, transcriptome, co-expression network, F730016J06Rik

## Abstract

Hematopoietic differentiation is a well-orchestrated process by many regulators such as transcription factor and long non-coding RNAs (lncRNAs). However, due to the large number of lncRNAs and the difficulty in determining their roles, the study of lncRNAs is a considerable challenge in hematopoietic differentiation. Here, through gene co-expression network analysis over RNA-seq data generated from representative types of mouse myeloid cells, we obtained a catalog of potential key lncRNAs in the context of mouse myeloid differentiation. Then, employing a widely used in vitro cell model, we screened a novel lncRNA, named Gdal1 (Granulocytic differentiation associated lncRNA 1), from this list and demonstrated that Gdal1 was required for granulocytic differentiation. Furthermore, knockdown of *Cebpe*, a principal transcription factor of granulocytic differentiation regulation, led to down-regulation of Gdal1, but not vice versa. In addition, expression of genes involved in myeloid differentiation and its regulation, such as *Cebpa*, were influenced in Gdal1 knockdown cells with differentiation blockage. We thus systematically identified myeloid differentiation associated lncRNAs and substantiated the identification by investigation of one of these lncRNAs on cellular phenotype and gene regulation levels. This study promotes our understanding of the regulation of myeloid differentiation and the characterization of roles of lncRNAs in hematopoietic system.

## 1. Introduction

Hematopoietic differentiation generates all cellular components of blood during hematopoiesis. As a main branch of the hematopoietic differentiation cascade, myeloid differentiation from hematopoietic stem cells (HSCs) give rise to various types of myeloid cells including common myeloid progenitors (CMPs), subsequently granulocyte–macrophage progenitors (GMPs), and finally their more mature progeny, granulocytes and monocytes/macrophages [1]. Recent studies have revealed that the hierarchical differentiation of blood cells is orchestrated by principal transcription factors (TFs) shaping gene regulatory networks in a lineage and differentiation-stage specific manner [2,3,4]. For example, the production of GMPs from CMPs is directed by C/EBPα [5,6], and C/EBPε is required for the generation of mature myeloid cells from GMPs [7,8]. Under this model of hematopoietic development, blockage in the myeloid differentiation at different stage could result in an accumulation of different immature myeloid cells in the bone marrow and peripheral blood, which is usually manifested by myeloid leukemia, leading to the concept that differentiation arrest is a causative event in malignant transformation [9]. The studies of acute myeloid leukemia (AML) mechanistically support this concept by demonstrating that cancer-associated differentiation arrest is caused by mutation or dysregulation of these differentiation-stage specific TFs [4]. A good example is, MLL is an essential TF to maintain HSCs [10], while chromosomal translocation involving MLL can induce myeloid differentiation arrest. The mouse bone marrow cells derived cell model expressing an inducible MLL fusion protein such as MLL-ENL has been widely used to study myeloid differentiation or malignant transformation by many research groups given its ease of manipulation [11,12,13,14].

Besides the orchestration of myeloid differentiation by TFs, a novel component of the gene regulatory network, lncRNAs, a subset of non-coding RNAs longer than 200 nt, without any apparent evidence of protein coding capacity, are also involved in myeloid differentiation. These include lnc-HSC, myeloid-specific HOTAIRM1, lnc-MC, lnc-DC, and lncRNA EGO [15,16,17,18,19,20]. In addition, many other lncRNAs have been reported to have mis-regulation and function during myeloid malignant transformation [21,22,23,24]. The last two decades of study on lncRNAs makes them less “dark” to us. The annotation of the mouse genome by GENCODE shows that there are more than 13,000 detected lncRNA genes [25]. The number of studies on lncRNAs has been increasing recent years, but the study of lncRNAs is still a considerable challenge given the large number of existing lncRNAs and the difficulty in experimentally exploring their roles. Furthermore, mechanisms of the participation of lncRNA in biological processes are not limited to transcription regulation but range from chromatin modulation to mRNA stability and protein activity regulation [26,27]. Thus, the addition of lncRNAs makes the gene regulatory network much more complicated in hematologic system [28]. A feasible approach to study lncRNAs would be to first narrow down the list of potential functional lncRNAs to help with further study of them in terms of a specific biological process. Fortunately, due to the application of high-throughput transcriptome sequencing technology in mounting studies in myeloid systems, it becomes possible to identify these lncRNAs by bioinformatics methods in an enormous and ever-growing amount of transcriptome sequencing data through bioinformatics methods for further study. Therefore, systematic identification of potential key lncRNAs in the context of myeloid differentiation is necessary to further study them and very critical to understand their roles in myeloid normal differentiation and malignant transformation.

In this study, we explored the involvement of lncRNAs in the context of mouse myeloid differentiation by gene co-expression network analysis. We obtained a catalog of potential key lncRNAs and validated the role of a novel lncRNA Gdal1 in myeloid differentiation. 

## 2. Materials and Methods

### 2.1. RNA-Seq Data Collection and Data Analysis

Transcriptome sequencing data of representative types of cells during myeloid differentiation were obtained from the Gene Expression Omnibus (GSE62734 and GSE73457) database and ArrayExpress database (E-MTAB-3079 and E-MTAB-2923). These types of cells were purified from mouse bone marrow as follows: c-Kit+ bone marrow cells (BMCs) were sorted by c-Kit+, HSCs were sorted by Lin−Sca1+ckit+, CMPs were sorted by Lin−CD127−Sca1−c-Kit+CD16/32 lowCD34+, precursor for myeloid-macrophage progenitors (PreGMPs) were sorted by Lin−Sca1−c-kit+CD150−CD105−CD41−CD16/32−, GMPs were sorted by Lin−CD127−Sca1−c-Kit+CD16/32highCD34+, granulocytes were sorted by Gr-1+Mac-1+CD3−CD4−CD8−IgM−Ter119−, monocytes were sorted by Gr-1mediumMac-1+CFMS+CD3−CD4−CD8−IgM−Ter119−, and macrophages were sorted by CD45+F4/80+CD31−Ter119−7AAD− [13,14,29].

The nascent RNA-seq data of a mouse model similar to CSH3 cells was obtained from ArrayExpress database (E-MTAB-3591).

FastQC was used to perform RNA-seq data quality control with the default parameters. Sequencing reads were mapped to GRCm39 by HISAT2 (v2.2.1) using default parameters with the GENCODE annotation vM26 as the reference transcriptome. StringTie (v2.1.5) was used to calculate the FPKM of a gene in each sample with the default parameters.

### 2.2. Differential Expression Analysis

Gene differential expression analysis was performed by DESeq2 for experiments with biological replicates and by edgeR for experiments without biological replicates. Genes with adjusted *p*-values ≤ 0.05 were considered to be differentially expressed.

### 2.3. Gene Co-Expression Network Analysis

Gene co-expression network analysis was performed using the R package WGCNA [30]. Genes with FPKM ≥ 0.1 in 25% or more of the samples were considered to be expressed in our dataset. Only mRNA and lncRNA genes were retained. Then, the FPKM was logarithmically transformed for subsequent gene co-expression network analysis by log (FPKM+1). In order to select the appropriate soft threshold power to fit the scale-free network model, we preset 25 soft powers from 1 to 25, and calculated the scale-free network fitting index corresponding to each one. The scale-free network fitting index exceeded 0.8 for the first time when the soft power was 19, so the gene expression adjacency matrix was calculated based on gene co-expression correlation coefficients for a signed network, with 19 chosen as the soft power. The topological overlay matrix (TOM) of the gene expression adjacency matrix was calculated and then converted into a dissTOM matrix by dissTOM = 1−TOM to facilitate clustering. Hierarchical clustering was performed on the dissTOM matrix, and the dynamic modules were cut from the cluster tree using the dynamic tree cut function implemented in the WGCNA R package. The minimum number of genes contained in a module was set to 30. The dynamic modules were merged with the correlation coefficient between their eigengenes ≥ 0.85 to obtain the final merged modules.

### 2.4. GO Enrichment Analysis

The GO terms enrichment analysis was performed online by DAVID 6.8, and the Fisher exact test was used to calculate the *p*-value to determine the significance of the enrichment.

### 2.5. Cell Culture

The CSH3 cell line was established by and gifted from the laboratory of Robert K. Slany [11]. CSH3 cells were cultured in RPMI-1640 (Gibco, Paisley, UK), supplemented with 10% heat-inactivated FBS (Gibco, Paisley, UK) and 1% penicillin/streptomycin (Gibco, Grand island, NY, USA) under 5% CO_2_ culture conditions, and 4 cytokines of mice, IL-6 (5 ng/mL, PeproTech, Rocky Hill, NJ, USA), IL-3 (5 ng/mL, PeproTech, Rocky Hill, NJ, USA), GM-CSF (5 ng/mL, PeproTech, Rocky Hill, NJ, USA), and SCF (50 ng/mL, PeproTech, Rocky Hill, NJ, USA) were required. Culture medium was replaced every 24 h. When 4HT (1 nM, Sigma, St. Louis, MI, USA) was added to the culture medium, the cells exhibited differentiation arrest and were transformed into myeloblastic cells, and CSH3 cells could differentiate toward mature granulocytes upon 4HT withdrawal.

The 32D cell lines were cultured in RPMI-1640 (Gibco, Paisley, UK), supplemented with 10% heat-inactivated FBS (Gibco, Paisley, UK), mouse IL-3 (10 ng/mL, Rocky Hill, NJ, USA), and 1% Penicillin/Streptomycin (Gibco, Grand island, NY, USA) under 5% CO_2_ culture conditions. For the induction of granulocyte differentiation, cells were washed with PBS twice and IL-3 was replaced by G-CSF (100 ng/mL, Rocky Hill, NJ, USA).

### 2.6. Total RNA Extraction

Total RNA was extracted using TRIzol reagent (Life Technologies, Carlsbad, CA, USA) according to the manufacturer’s instructions.

### 2.7. Reverse Transcription and Real-Time qPCR

The cDNA was synthesized with Reverse Transcription System (Promega, Madison, WI, USA) according to the manufacturer’s protocol. Real-time qPCR analysis was performed using SYBR Green qPCR Master Mix (Thermo Scientific, Waltham, MA, USA). The relative expression of a gene was normalized using the 2−ΔΔCt method with *Gapdh* as the internal control except specific statement, and two-tailed t-test was used to determine the significance of expression differences. All reactions were run on a CFX96 TOUCH Real Time PCR System (Bio-Rad, Hercules, CA, USA) with triplicates. All PCR primers are listed in Appendix A.

### 2.8. RNA-Seq

The total RNAs were extracted from CSH3 cells according to the standard protocol. RNA quality control, RNA-seq library preparation, and high through-put sequencing were performed by Novogene Co., Ltd. in Beijing, China. Poly-A RNAs were enriched for the preparation of RNA-seq library and reads were generated from Illumina Hi-seq sequencing platforms.

### 2.9. Gene Knockdown Using ASOs

For knockdown of Gdal1, ASOs for isoform 1 and isoform 2 of Gdal1 were designed and synthesized since these two isoforms are the main transcripts of Gdal1 in CSH3 cells and were dissolved in sterile water to a final concentration of 50 μM. Mixture of these two ASOs in equal proportions was electro-transfected into CSH3 cells at a working concentration of 2 μM, and negative control ASOs was electro-transfected into CSH3 cells at the same concentration. Upon withdrawal of 4HT, ASOs electro-transfection was performed every 24 h to achieve continuous interference with Gdal1. The experiments were carried out according to the manual of the Amaxa Cell Line Nucleofector Kit L (Lonza, Koln, Germany) using Lonza Amaxa Nucleofector 2b with the electro-transfection program X-001. The sequences of ASOs for isoform 1 is (5′→3′): TGAAGCAGACACATGAGCGG, and the sequences of ASOs for isoform 2 is (5′→3′): CTGACACGCTGATACACCTA. The control ASOs were provided by GUANGZHOU RIBOBIO Co., Ltd (Guangzhou, China).

### 2.10. CRISPRi

CRISPRi was carried out as previously described with some modifications [31]. Briefly, based on the annotations of the mouse gene transcription start site (TSS) in the FANTOM5 database, gRNAs were designed near TSS in the genomic region [−50 bp, 300 bp], the design tool was http://crispr.mit.edu/, accessed on 21 May 2018. We selected the gRNAs with higher scores. For *Cebpe*, FANTOM5 has annotated two TSSs that were far from each other, we designed one gRNA for each of the 2 TSSs, respectively, and then used the dual-gene CRISPRi gene silencing system to express them in CSH3 cells simultaneously for the knockdown of *Cebpe*. For Gdal1, we designed a total of 6 gRNA sequences, and then used the single-gene CRISPRi gene silencing system to knockdown the expression of Gdal1. In addition, we also synthesized a negative control gRNA. The sequences of all designed gRNAs are listed in Appendix A. Lentiviral particles were generated and harvested from HEK293T cells, then transfected into CSH3 cells. GFP positive CSH3 cells were sorted on FACSAria II flow cytometer (BD Biosciences, San Jose, CA, USA).

### 2.11. Flow Cytometry Analysis

Cells were washed twice with cold phosphate-buffered saline (PBS) and re-suspended in 100 μL of cold PBS, then incubated for 30 min on ice in the dark with Gr-1 and Mac-1 antibody and isotype control. After that, cells were washed twice with cold PBS and re-suspended in 300 μL of cold PBS and analyzed on FACSCalibur flow cytometry (BD Biosciences, San Jose, CA, USA). Flow data analysis was performed using FlowJo.

### 2.12. Western Blotting

Cells were harvested for whole protein collection. Protein samples were separated on SDS-polyacrylamide gel and transferred to Polyvinylidene Fluoride Membranes (Millipore). Membranes were incubated with primary antibodies against CEBPE and β-actin, then HRP-labeled secondary antibody. Chemiluminescent signals were detected using ECL Prime Western blotting Detection Reagent (GE Healthcare, Chicago, IL, USA), and membranes were imaged using an automatic imaging analysis system (Tanon 5200, Shanghai, China).

### 2.13. Antibodies Used for Flow Cytometry Analysis and Western Blot

Anti-mouse Ly-6G (Gr-1) APC (eBioscience, San Diego, CA, USA), Anti-mouse CD11b (Mac-1) PE (eBioscience, San Diego, CA, USA), Rat IgG2a K Isotype PE (eBioscience, San Diego, CA, USA), Rat IgG2a K Isotype APC (eBioscience, San Diego, CA, USA), C/EBP epsilon antibody (GeneTex, Irvine, CA, USA), and β-actin antibody (Abcam, Cambridge, UK).

## 3. Results

### 3.1. Gene Co-Expression Network Analysis and Potential Key lncRNAs in the Context of Myeloid Differentiation

Transcriptome sequencing data of representative types of cells in the context of mouse myeloid differentiation were collected. Gene expression data from mouse myeloid cells including c-Kit+ bone marrow cells (BMCs), HSCs, CMPs, PreGMPs, GMPs, granulocytes, monocytes, and macrophages, simulated gene expression dynamics of myeloid differentiation (Figure 1a). In addition, as some stem/progenitor cells expressing inducible MLL-ENL fusion proteins represent cellular state of differentiation arrest, transcriptome sequencing data of these cells were also included in the following bioinformatics analysis (Figure 1a). We finally obtained data from 25 samples and the original cells generating these data were all sorted from BMCs of the C57BL/6 mouse strain by flow cytometer. These data were all based on Illumina Hiseq 2000 platform and cDNA libraries were all prepared with poly-A RNA enriched from total RNA (Appendix A). Sequencing reads were mapped to the mouse GRCm39 and the mapping rates of all samples were above 80% (Appendix A). The FPKM was used to measure the expression level of a gene in a sample. If the FPKM of a gene was ≥0.1 in 25% or more of the samples, this gene was thought to be expressed in our data set. Retaining only the lncRNA and mRNA, a total of 17,061 genes were expressed in our data set, including 3257 lncRNAs and 13,804 mRNAs.

To identify potential key lncRNAs, these expressed genes were grouped into functional modules through gene co-expression network analysis using the WGCNA algorithm [30]. In a gene co-expression network, genes with higher expression correlation tend to be clustered into the same module, implying their involvement in related biological pathways. These 17,061 genes formed a total of 32 modules. The smallest module contained 49 genes, and the largest contained 3390 genes (Figure 1b and Appendix A). There were 5 genes not grouped into any functional module due to their bad expression correlations with other genes, and they were put into a group named module grey. According to network theory, the degree of a node refers to the number of nodes connected to it. The higher the degree of a gene, the more genes with similar expression patterns to it and the more essential it is in this module. Then, the distribution of intra-module degree of all lncRNAs and mRNAs were analyzed separately. The distribution density of lncRNA was lower than that of mRNA when the intra-module degree was high, and it was higher when the intra-module degree was very low (Figure 1c).

We defined those genes with an intra-module degree ranking top 25% in each module in the gene co-expression network as essential genes in the context of myeloid differentiation. Then, the essential genes were counted and a total of 2042 genes were obtained including 1862 coding genes and 180 lncRNA genes, accounting for about 13.5% of the total expressed coding genes and 5.5% of total expressed lncRNAs, respectively (Figure 1d and Appendix A). Among these lncRNAs, Gm35551 was previously identified as lnc-HSC-1 to control in vitro and in vivo differentiation of HSC [15]. SNHG3 was shown to be up-regulated in acute myeloid leukemia (AML) samples and cells, and knockdown of SNHG3 could inhibit leukemia cell proliferation and induce cell apoptosis [32]. Neat1 was required for ATRA-induced granulocytes differentiation [24]. These reports indicated that the gene co-expression network analysis and our screen criteria could discover the essential lncRNAs. Next, the distributions of these essential lncRNA and mRNA genes across all chromosomes were analyzed separately (Figure 1e). We spotted that the essential lncRNAs on chromosome 1–11 accounted for about 72.2% of all the essential lncRNAs and the essential mRNAs on chromosome 1–11 accounted for 66% of all the essential mRNAs, while the size of chromosome 1–11 accounted for only 60.5% of the whole genome.

### 3.2. Gdal1 Was Required for Granulocytic Differentiation

We employed a widely used mouse cell model, CSH3 cells, to verify that we effectively identified the potential key lncRNAs in myeloid differentiation. CSH3 cells are derived from mouse c-Kit+ BMCs, and 4-hydroxy-tamoxifen (4HT) can induce the activation of the MLL-ENL fusion protein in CSH3 cells. When 4HT is absent in the culture medium, CSH3 cells can differentiate toward mature granulocytes. This differentiation can be reversibly blocked by MLL-ENL activation with the presence of 4HT [11]. This cell model allows us to simulate cellular states of myeloid differentiation and differentiation arrest in vitro.

To select candidate lncRNAs for validation from the identified potential key lncRNAs, RNA-seq analyses were performed in CSH3 cells in normal differentiation (3 days after 4HT withdrawal) or differentiation arrest state (presence of 4HT) to obtain the lncRNAs expression profiles. The expression levels of most lncRNAs were similar between these two states, while 72 lncRNAs were significantly highly expressed in differentiating CSH3 cells, the other 122 lncRNAs in differentiation arrest cells (Figure 2a). In consideration of the identified essential lncRNAs by gene co-expression network analysis above, we further narrowed down the scope of lncRNAs by overlapping the 194 differentially expressed lncRNAs with the 180 essential lncRNAs and obtained 14 lncRNAs, of which 4 were highly expressed in differentiated CSH3 cells and the other 10 were highly expressed in differentiation arrest cells (Table 1). These 14 lncRNAs were considered to be important in the context of granulocytic differentiation of CSH3 cells. 

Among these 14 lncRNAs, we focused on a novel lncRNA, F730016J06Rik, hereinafter referred to as Gdal1. Gdal1 was enriched in differentiated CSH3 cells, and its expression level in differentiated cells was about 5.5 times higher than that in arrested cells according to RNA-seq data. More importantly, in the gene co-expression network, Gdal1 and two important myeloid differentiation cell surface marker genes, Gr-1 and Mac-1, were grouped into the same functional module (Appendix A), indicating a high expression correlation between Gdal1 and them and its association to myeloid differentiation. Furthermore, based on the gene expression profiles of representative types of cells during mouse myeloid differentiation generated by Qian et al. [29], Gdal1 was significantly highly expressed in most mature myeloid cells except dendritic cells, while its expression level was very low in various hematopoietic stem/progenitor cells such as HSC, CMP, and GMP (Figure 2b), suggesting that its expression is strictly regulated during myeloid differentiation and Gdal1 is specifically expressed in mature myeloid cells. Based on these data above, we hypothesized that Gdal1 was required for granulocytic differentiation of CSH3 cells.

Subsequently, the expression dynamics of Gdal1 during granulocytic differentiation of CSH3 cells was tested. Mouse Gdal1 gene with three different isoforms is located on chromosome 2 (Figure 2c). The expression of the three isoforms of Gdal1 were measured in CSH3 cells by the qPCR method, and isoform 2 was the dominant transcript of Gdal1 accounting for 80% of the total expression level, followed by isoform 1 accounting for about 15%, and the expression level of isoform 3 was the lowest accounting for only about 5% (Figure 2d). In subsequent research, the expression level of Gdal1 was measured as the sum of isoform 2 and isoform 1. Then, we found that the expression of Gdal1 began to rise within 2 days after the initiation of the differentiation of CSH3 cells, and increased sharply in the following 2 days, then maintained at a high level (Figure 2e). These data showed that the expression of Gdal1 increased with the differentiation of CSH3 cells.

In order to verify the requirement of Gdal1 in the process of granulocyte differentiation, we then interfered with the expression of Gdal1 and evaluated its influence on granulocytic differentiation of the CSH3 cell. Two antisense oligonucleotides (ASOs) were designed and electro-transfected into differentiating CSH3 cells every 24 h to knockdown Gdal1 continuously, and the expression of Gdal1 and the differentiation state of CSH3 cells were tested every 24 h. At each time point after the differentiation of CSH3 cells, Gdal1 was knocked down to less than 80% of the control cells (Figure 2f). Flow cytometry analysis of two myeloid differentiation cell surface markers, Gr-1 and Mac-1, showed that Mac-1 was down-regulated in Gdal1 knockdown cells 4 days after the initiation of the differentiation compared with control cells, revealing that the differentiation of CSH3 cells was inhibited by Gdal1 knockdown, and similar phenomena could still be observed on the 5th day (Figure 2g). In addition to transient gene silencing with ASOs, we also knocked down Gdal1 stably by CRISPRi and evaluated its influence on granulocytic differentiation of CSH3 cells. Six gRNAs targeting the promoter region of Gdal1 were designed. gRNA2 was the most effective, reducing the RNA level of Gdal1 by about 60% (Figure 2h). Then, gRNA2 was used to knockdown Gdal1 in subsequent research. Flow cytometry analysis of two myeloid differentiation cell surface markers, Gr-1 and Mac-1, revealed that knockdown of Gdal1 by CRISPRi could also lead to the inhibition of CSH3 cells differentiation (Figure 2i). Compared with control cells, Gdal1 knockdown cells showed differentiation inhibition on the 5th day after the initiation of the differentiation, and the inhibition was the most obvious on the 6th day. These data demonstrated that the inhibition of Gdal1 expression could cause the blockage of the granulocytic differentiation of CSH3 cells.

Furthermore, we also verified the requirement of Gdal1 for granulocyte differentiation in 32D cells. The expression of Gdal1 was up-regulated during granulocyte differentiation of 32D cells induced by G-CSF (Figure 2j). Then, Gdal1 was knocked down stably by CRISPRi (Figure 2k), and the blockage of granulocyte differentiation of 32D cells was also observed (Figure 2l). The above results showed that Gdal1 was specifically overexpressed in mature myeloid cells, and the expression of Gdal1 increased with the granulocytic differentiation of CSH3 cells and 32D cells. Knockdown of Gdal1 led to inhibition of granulocytic differentiation of CSH3 cells and 32D cells. It suggests that Gdal1 is required for granulocyte differentiation.

### 3.3. Knockdown of Cebpe Led to Down-Regulation of Gdal1, but Not Vice Versa

Next, we tried to further confirm the involvement of Gdal1 in granulocytic differentiation at the level of gene regulation. We first revealed which genes can influence the expression of Gdal1. According to the data generated by nascent RNA-seq at 0, 8, 16, 24, 48, and 72 h after the initiation of myeloid differentiation [12], we plotted the synthesis rate of Gdal1 and found it did not change significantly within 24 h, but increased significantly in 48 h and thereafter (Figure 3a), which was consistent with the phenomenon we observed in CSH3 cells before. This suggested that the genes with significant changes in the rate of RNA synthesis within 24 h or 48 h after differentiation may affect the expression of Gdal1. Therefore, we analyzed the gene expression differences between 24 h and 0 h as well as 48 h and 0 h by using nascent RNA-seq data to identify these genes. RNA synthesis rates of 319 genes changed significantly within 24 h and 1622 genes within 48 h after the initiation of differentiation. Then, we obtained a total of 1717 genes by combining these two groups of genes (Figure 3b), which were believed to contain genes that can influence the expression of Gdal1. Considering that Gdal1 is a regulator of granulocytic differentiation, we overlapped these 1717 genes with all the 421 genes annotated by the Gene Ontology (GO) consortium that are directly related to myeloid differentiation and its regulation to narrow down the candidate list (Figure 3c), resulting in 50 genes, including 19 TFs and 31 other coding genes (Appendix A).

Since TFs play very critical regulatory roles during hematopoietic differentiation, we focused on these 19 TFs to find the upstream regulator of Gdal1. Based on the nascent RNA-seq data, gene expression correlation analyses were performed between the 19 TFs and Gdal1 (Table 2). Expression of *Cebpe* strongly positively correlated with the expression of Gdal1 (cor. ≥ 0.95), indicating its association to the expression of Gdal1. Meanwhile, the expression of another 5 TF genes, *Meis1*, *Mef2c*, *Nkx2-3*, *Sox4*, and *Hoxa7* had a strong negative correlation with the expression of Gdal1 (cor. ≤ −0.95). Since *Meis1* and *Hoxa7* are highly expressed in stem/progenitor cells and contribute to myeloid differentiation blockage, it implied the roles of Gdal1 in cell differentiation from another perspective. *Cebpe* is highly expressed and required in late stage of myeloid differentiation, especially granulocyte differentiation [33,34]. The expression of *Cebpe* gradually increased upon withdrawal of 4HT as CSH3 cells began to differentiate toward mature granulocytes (Figure 3d,e). We thus inferred that the expression of Gdal1 may be influenced by *Cebpe*.

To verify the inference that the expression of Gdal1 is influenced by *Cebpe*, two gRNAs were designed to knockdown *Cebpe* stably using two-gene CRISPRi system in CSH3 cells. We found that knockdown of *Cebpe* not only led to the decrease of Gdal1 expression when the cells were in differentiation state (3 days after 4HT withdrawal) but also when the cells were in differentiation arrest state (presence of 4HT) (Figure 3f,g). Furthermore, we also detected whether the expression of *Cebpe* was influenced by Gdal1. The expression of *Cebpe* in Gdal1 knocked down cells was checked at the RNA level, and it was found that *Cebpe* was not affected by Gdal1 knockdown whether the cells were in differentiation (3 days after 4HT withdrawal) or differentiation arrest state (presence of 4HT) (Figure 3h). These data indicated that as a principal TF for granulocyte differentiation, *Cebpe* could influence the expression of Gdal1, but not vice versa.

### 3.4. Transcriptome Analysis in Gdal1 Knockdown CSH3 Cells Further Confirmed the Involvement of Gdal1 in Myeloid Differentiation

Finally, we confirmed the necessity of Gdal1 in the process of granulocyte differentiation by detecting the genes affected in Gdal1 knockdown CSH3 cells with blocked differentiation. For this purpose, we sequenced the transcriptome of control group and Gdal1 knockdown CSH3 cells 6 days after the initiation of differentiation, which was the time when the phenotype of differentiation inhibition was most obvious. Two pairs of knockdown/control cells were collected to generate transcriptome sequencing data. Each library produced more than 10 Gb of high-quality reads, and the mapping rates were all above 90%. A total of 372 differentially expressed genes were identified (Figure 4a, Appendix A). Among these genes, *Cebpa* and *Irf8* were increased in Gdal1 knockdown CSH3 cells with differentiation blockage. *Cebpa* is one of principal TFs orchestrating myeloid differentiation, and precise control of its expression is essential for normal myeloid differentiation [35,36]. Another two genes related to myeloid differentiation, *Clec5a* and *Jak2* were down-regulated in Gdal1 knockdown CSH3 cells with differentiation blockage. CLEC5A is a cell surface receptor involved in the activation of myeloid cells and its expression is associated with granulocytic differentiation of 32Dcl3 cells [37,38]. In addition, GO terms enrichment analysis showed that some differentially expressed genes were involved in myeloid differentiation regulatory pathways such as ERK cascade and NF-kappaB signaling (Figure 4b, Appendix A). Moreover, the expression of *Pld4*, *Dock7*, *Rest*, *Inhba* and *Ssbp3*, which were annotated to be involved in differentiation of hematopoietic progenitor cells were also influenced in Gdal1 knockdown CSH3 cells with differentiation blockage (Figure 4b, Appendix A).

Among these genes, we paid attention to another famous C/EBP family member *Cebpa*. *Cebpa* is usually highly expressed during the differentiation from CMPs to GMPs [6], and then decreases to relative low level during the subsequent granulocyte differentiation [39]. We detected the expression of *Cebpa* during the granulocytic differentiation of control and Gdal1 knockdown CSH3 cells. The expression of *Cebpa* remained low after the differentiation of control cells but increased on the 5th and 6th day after the differentiation of Gdal1 knockdown cells (Figure 4c). These results suggested that in differentiating CSH3 cells, if Gdal1 was knocked down, the expression of *Cebpa* could return to a higher level. Furthermore, since *Cebpe* is required for granulocyte differentiation and knockdown of *Cebpe* can lead to down-regulation of Gdal1. The expression of *Cebpa* should increase in *Cebpe* knockdown CSH3 cells, and this was validated by qPCR (Figure 4d). Thus, in Gdal1 knockdown CSH3 cells with differentiation blockage, expression of genes related to myeloid differentiation and its regulation were changed, further confirming the important role of Gdal1 in the process of myeloid differentiation.

Moreover, we also investigated gene expression patterns of these 3 genes in representative cell types during myeloid differentiation in BloodSpot database [40]. As we could see, expression level of *Cebpe* and Gdal1 were extremely low in myeloid progenitors, but sharply up-regulated in granulocytes, while *Cebpa* was highly expressed in CMPs and GMPs, and its expression level became very low in granulocytes (Figure 4e). These expression patterns were consistent with our finding that *Cebpe* could positively affect expression of Gdal1, and *Cebpa* expression was up-regulated in Gdal1 knockdown CSH3 cells with differentiation blockage. In addition, granulocytes and macrophages showed a similar change trend of these 3 genes during final differentiation from GMP to mature cells, whereas the expression of *Cebpe* and Gdal1 did not increase obviously during final differentiation and the expression of *Cebpa* was just slightly lower in monocytes compared to GMP (Figure 4e). This result implied the relationship among the 3 genes.

## 4. Discussion

In this study, we narrowed down the lncRNAs list in the context of mouse myeloid differentiation by producing a catalog of potential essential lncRNAs based on intramodule degree by gene co-expression network analysis. This result is an important resource for further biological research of lncRNAs in normal myeloid differentiation and also in myeloid transformation. From the potential key lncRNA catalog we identified, we then demonstrated that a new lncRNA Gdal1 enriched in differentiated cells was important for granulocyte differentiation. This reflects the effectiveness of our research strategy of combining gene co-expression network analysis and differential expression analysis to find interested lncRNAs, and it could be expanded to the study of lncRNAs with unknown functions in other biological systems. Meanwhile, the involvement of other potential key lncRNAs in the process of myeloid differentiation is worthy of further experimental confirmation. 

After we identified the role of Gdal1 in myeloid differentiation, we further studied the regulation of Gdal1. We found that *Cebpe* could positively regulate the expression of Gdal1 in CSH3 cells whether they were in the state of normal differentiation or arrested differentiation. *Cebpe* is a famous C/EBP family member that is required and highly expressed during granulocyte terminal differentiation. However, the specific mechanism of *Cebpe* controlling the differentiation process remains unclear. The finding of targeting Gdal1 may give some clues for understanding the mechanisms of *Cebpe* and introduce a novel non-coding regulator to this process. 

In addition, we also observed that the expression of genes related myeloid differentiation was affected in Gdal1 knockdown CSH3 cells with blocked differentiation. These genes contain *Cebpa* and genes involved in the ERK signaling pathway (Figure 4b, supplementary Appendix A). The ERK signaling pathway plays an important role in myeloid differentiation through directly inhibiting the activity of CEBPA protein by phosphorylation of the 21st serine of CEBPA protein [41]. Therefore, we could speculate that the function of *Cebpa* may be affected by the ERK pathway in Gdal1 knockdown CSH3 cells with blocked differentiation. Although the expression of many genes has changed after Gdal1 knockdown, it is not clear whether these genes are the direct target genes of Gdal1. We transiently knocked down Gdal1 by ASOs in differentiating CSH3 cells but did not observe significant up-regulation of *Cebpa* at 24 h. It suggests that *Cebpa* might not be the direct target gene of Gdal1. However, *Cebpa* is still one of the genes whose expression level changed significantly after Gdal1 knockdown, which may be indirectly regulated by Gdal1. To identify the direct target gene of a lncRNA, ChIRP (chromatin isolation by RNA purification) or RAP (RNA antisense purification) techniques would be advised to pull down the binding target sequence of the lncRNA. Therefore, the direct and indirect effect of Gdal1 knockdown needs further experimental verification.

Since Gdal1 has an important function in mouse myeloid differentiation, we examined the conservation of Gdal1. We checked the multiple genome alignments of 35 vertebrate species in the UCSC genome browser and found that genomic region of Gdal1 is conserved between mice, rats, and Chinese hamsters, but far less conserved between mice and the remaining species including Euarchontoglires and other Glires such as beavers, guinea pigs, and squirrels. We also searched for transcripts with similar primary sequences to Gdal1 in humans using BLAST and no significant similarity was found. These results indicate that Gdal1 is not conservative at the level of primary sequence, but it doesn’t rule out that the secondary structure or short conserved elements is conservative. The conservation of Gdal1 in Muridae suggests that lncRNA has species specificity, but there still may exist a counterpart in humans. All these are worthy of further study.

In summary, this study is of significance to understand the regulation of myeloid differentiation and the role of lncRNAs in hematopoietic system and provides new insight for the study of a large number of lncRNAs with unknown functions.

## 5. Conclusions

In conclusion, we systematically identified myeloid differentiation associated lncRNAs and demonstrated that one of them, named Gdal1 was required for granulocytic differentiation. *Cebpe* and *Cebpa* were involved in the regulatory network of Gdal1. This study promotes our understanding of the regulation of myeloid differentiation and the characterization of roles of lncRNAs in hematopoietic system.

## Figures and Tables

**Figure 1 genes-12-00630-f001:**
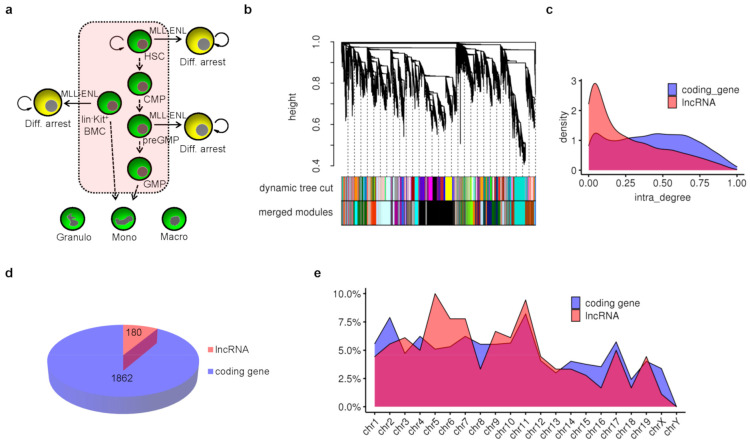
Potential key lncRNAs in the context of mouse myeloid differentiation. (**a**) All cell types involved in gene co-expression network analysis and their hierarchical differentiation relationship were shown. Diff. is short for differentiation. (**b**) Cluster tree and gene modules in gene co-expression network analysis. All genes were clustered into dynamic modules based on gene expression correlation, and dynamic modules with strong correlation between their eigengenes were merged to form final merged gene modules in gene co-expression network. (**c**) Distribution of intra-module degree for mRNA and lncRNA genes. (**d**) Total number of potential key mRNA and lncRNA genes with intra-module degree ranking top 25% in each module. (**e**) Distribution of these potential key lncRNA and mRNA genes across all chromosomes.

**Figure 2 genes-12-00630-f002:**
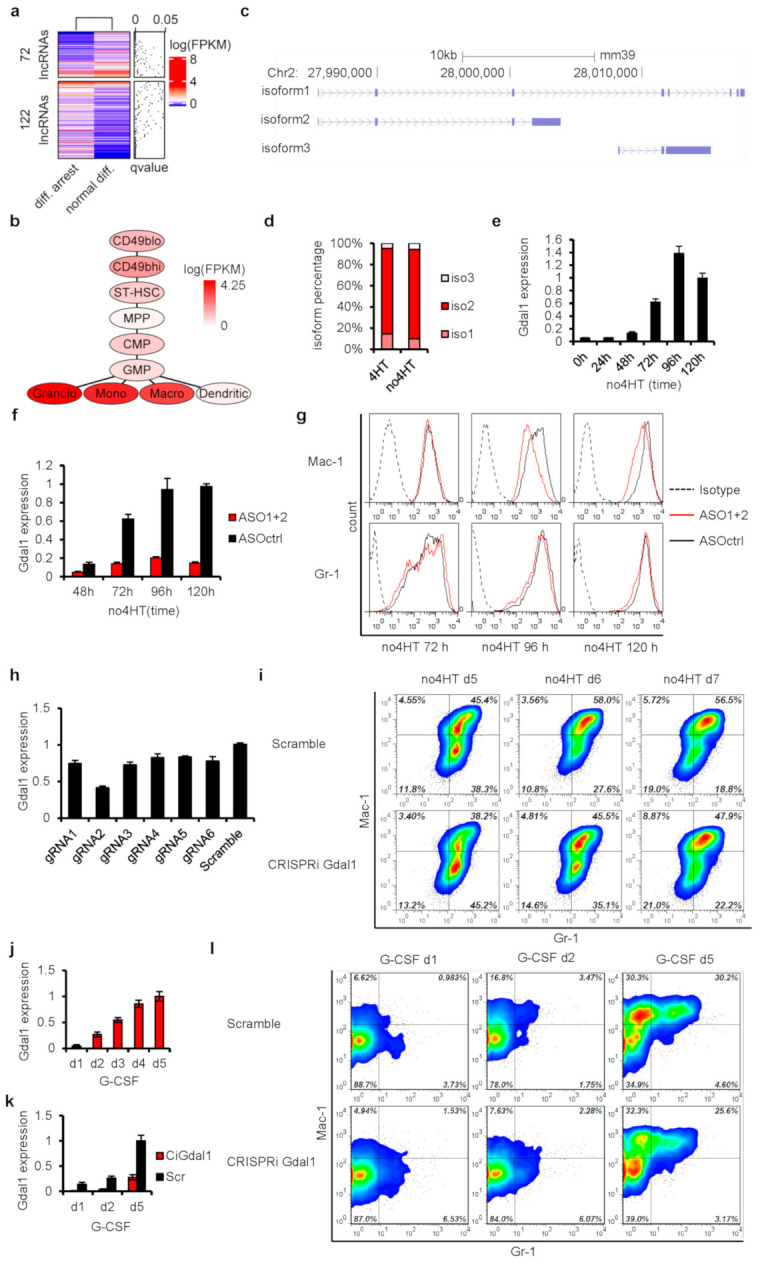
Gdal1 was required for granulocytic differentiation. (**a**) The 72 lncRNAs and 122 lncRNAs enriched in differentiating and differentiation arrest CSH3 cells, respectively, were shown in heat-map. (**b**) Expression dynamics of Gdal1 during myeloid differentiation. CD49blo and CD49bhi stand for low and high CD49b HSCs, respectively; ST-HSC is short time HSCs. (**c**) GENCODE annotation of 3 isoforms of Gdal1. (**d**) Expression proportion of the 3 isoforms of Gdal1 in differentiating (3 days after 4HT withdrawal) and differentiation arrest (with the presence of 4HT) CSH3 cells. Relative expression was analyzed by qPCR, data were shown as mean of 3 replicates. (**e**) Dynamics of Gdal1 expression during granulocytic differentiation of CSH3 cells. Relative expression of Gdal1 was analyzed by qPCR. (**f**) Knockdown of Gdal1 by ASOs was tested by qPCR. (**g**) Knockdown of Gdal1 by ASOs inhibited granulocytic differentiation of CSH3 cells. Granulocytic differentiation of CSH3 was assessed by flow cytometry analyses of myeloid differentiation cell surface marker Gr-1 and Mac-1. (**h**) Knockdown of Gdal1 by CRISPRi using 6 gRNAs. Relative RNA level of Gdal1 was measured by qPCR, and gRNA2 was the most effective. (**i**) Knockdown of Gdal1 inhibited granulocytic differentiation of CSH3 cells. CSH3 cells were gated by Gr-1+/CD11b+, and the changes of Gr-1 and CD11b were tracked in control cells and Gdal1 knockdown cells according to the time. Contour plots showed the results analyzed by flow cytometry. Red color stands for high cell density and blue for low density. (**j**). Dynamics of Gdal1 expression during granulocytic differentiation of 32D cells induced by G-CSF. Relative expression of Gdal1 was analyzed by qPCR. (**k**) Knockdown of Gdal1 by CRISPRi in G-CSF induced 32D cells. (**l**) Knockdown of Gdal1 inhibited G-CSF induced granulocytic differentiation of 32D cells. 32D cells were gated by Gr-1+/CD11b+, and the changes of Gr-1 and CD11b were tracked in control cells and Gdal1 knockdown cells according to the time. Contour plots showed the results analyzed by flow cytometry. Red color stands for high cell density and blue for low density. qPCR data is represented as mean ± SEM of 3 replicates.

**Figure 3 genes-12-00630-f003:**
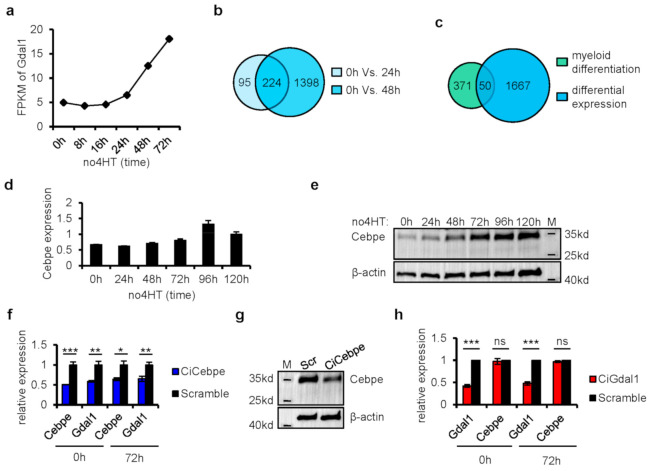
Knockdown of *Cebpe* led to down-regulation of Gdal1, but not vice versa. (**a**) Synthesis rate of Gdal1 is represented as FPKM of nascent Gdal1 RNA after the initiation of granulocyte differentiation. (**b**) Genes with changes of synthesis rate at 24 h and 48 h after the initiation of myeloid differentiation compared with 0 h. (**c**) Overlapping the differentially expressed genes shown in (**b**) with all the 421 genes annotated by Gene Ontology (GO) consortium that are directly related to myeloid differentiation. (**d**,**e**) Gene expression dynamics of *Cebpe* in RNA level by qPCR and in protein level by Western blotting during the differentiation of CSH3 cells. (**f**) Knockdown of *Cebpe* by CRISPRi resulted in down-regulation of Gdal1 in both differentiating (3 days after 4HT withdrawal) and differentiation arrest (with the presence of 4HT) CSH3 cells. Relative RNA level was analyzed by qPCR. (**g**) Knockdown of *Cebpe* by CRISPRi was confirmed in protein level in differentiating CSH3 cells by Western blotting. (**h**) Knockdown of Gdal1 did not affect the expression of *Cebpe* in both differentiating (3 days after 4HT withdrawal) and differentiation arrest (with the presence of 4HT) CSH3 cells. RNA levels of genes are analyzed by qPCR. Ci is short for CRISPRi, qPCR data is represented as mean ± SEM of 3 replicates. ns *p* > 0.05, * *p* ≤ 0.05, ** *p* ≤ 0.01, *** *p* ≤ 0.001.

**Figure 4 genes-12-00630-f004:**
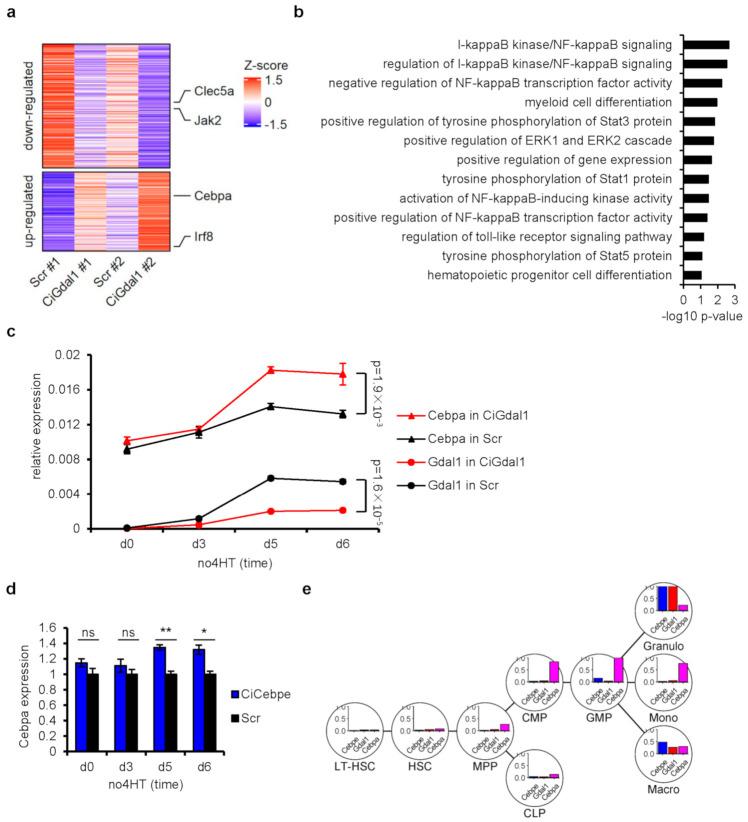
Transcriptome analysis in Gdal1 knockdown CSH3 cells further confirmed the involvement of Gdal1 in myeloid differentiation. (**a**) Genes influenced in Gdal1 knockdown CSH3 cells with differentiation blockage. The heat map shows the expression level of 372 genes based on transcriptome sequencing in two pairs of knockdown/control cells. Up-panel shows down-regulated genes in Gdal1 knockdown cells, and low-panel shows up-regulated genes in Gdal1 knockdown cells. Marked genes are related to myeloid differentiation. (**b**) GO term enrichment analysis of influenced genes in Gdal1 knockdown cells. Biological process GO terms related to myeloid differentiation and its regulation were shown. (**c**) The expression of *Cebpa* was up-regulated in Gdal1 knockdown CSH3 cells with differentiation blockage. Relative expressions of genes were analyzed by qPCR in Gdal1 knockdown and control differentiating CSH3 cells. 2^−ΔCt^ is used to data normalization. (**d**) The expression of *Cebpa* was also up-regulated in *Cebpe* knockdown CSH3 cells. RNA level of *Cebpa* was analyzed by qPCR. The expression of *Cebpa* increased significantly on the 5th and 6th day after the initiation of differentiation. (**e**) Gene expression patterns of *Cebpe*, Gdal1, and *Cebpa* during myeloid differentiation in BloodSpot database. Gene expression levels of each gene normalized to its maximal value in these cell types are shown. Ci stands for CRISPRi. Gene expression data by qPCR is represented as mean ± SEM of 3 replicates. ns *p* > 0.05, * *p* ≤ 0.05, ** *p* ≤ 0.01.

**Table 1 genes-12-00630-t001:** Information of 14 important lncRNA genes in the context of CSH3 cells differentiation.

Ensemble ID	Gene Name	Location	Scaled Degree	Enriched In ^1^	*p*-Value
ENSMUSG00000118061	Rbfaos	chr18	0.771	Diff. arrest	8.45 × 10^−13^
ENSMUSG00000104184	Gm37818	chr11	0.833	Diff. arrest	1.17 × 10^−11^
ENSMUSG00000086425	Gdal1	chr2	0.754	Normal Diff.	1.73 × 10^−8^
ENSMUSG00000084085	Gm16140	chr11	0.751	Diff. arrest	2.77 × 10^−5^
ENSMUSG00000109799	Gm45515	chr7	0.821	Diff. arrest	4.23 × 10^−5^
ENSMUSG00000027196	Alkbh3os1	chr2	0.834	Diff. arrest	6.86 × 10^−5^
ENSMUSG00000087026	A230103J11Rik	chr8	0.772	Diff. arrest	3.00 × 10^−4^
ENSMUSG00000097057	Gm17638	chr15	0.844	Diff. arrest	5.28 × 10^−4^
ENSMUSG00000107320	Gm42549	chr6	0.780	Diff. arrest	4.61 × 10^−3^
ENSMUSG00000107480	Gm44165	chr7	0.757	Normal Diff.	8.90 × 10^−3^
ENSMUSG00000113184	Gm49654	chr12	0.836	Normal Diff.	1.75 × 10^−2^
ENSMUSG00000108402	9430064I24Rik	chr7	0.817	Diff. arrest	2.28 × 10^−2^
ENSMUSG00000078308	Gm47854	chr9	0.905	Normal Diff.	3.85 × 10^−2^
ENSMUSG00000097772	5430416N02Rik	chr5	0.960	Diff. arrest	3.85 × 10^−2^

^1^ Diff. is short for differentiation.

**Table 2 genes-12-00630-t002:** Nineteen TFs that potentially influence the expression of Gdal1.

Ensemble ID	Gene Name	Correlation Coefficient	*p*-Value (*t*-Test)
ENSMUSG00000086425	Gdal1	1.000	0.00
ENSMUSG00000052435	Cebpe	0.991	1.12 × 10^−4^
ENSMUSG00000020160	Meis1	−0.991	1.22 × 10^−4^
ENSMUSG00000005583	Mef2c	−0.964	1.96 × 10^−3^
ENSMUSG00000044220	Nkx2-3	−0.962	2.13 × 10^−3^
ENSMUSG00000076431	Sox4	−0.961	2.26 × 10^−3^
ENSMUSG00000038236	Hoxa7	−0.953	3.29 × 10^−3^
ENSMUSG00000055148	Klf2	0.947	4.08 × 10^−3^
ENSMUSG00000022508	Bcl6	0.946	4.37 × 10^−3^
ENSMUSG00000038227	Hoxa9	−0.941	5.17 × 10^−3^
ENSMUSG00000034957	Cebpa	0.939	5.39 × 10^−3^
ENSMUSG00000021025	Nfkbia	0.935	6.15 × 10^−3^
ENSMUSG00000021356	Irf4	0.933	6.62 × 10^−3^
ENSMUSG00000038253	Hoxa5	−0.896	1.55 × 10^−2^
ENSMUSG00000041515	Irf8	−0.875	2.25 × 10^−2^
ENSMUSG00000020644	Id2	0.814	4.87 × 10^−2^
ENSMUSG00000037465	Klf10	0.784	6.48 × 10^−2^
ENSMUSG00000015053	Gata2	0.645	1.66 × 10^−1^
ENSMUSG00000052684	Jun	0.507	3.05 × 10^−1^
ENSMUSG00000022528	Hes1	0.293	5.73 × 10^−1^

## Data Availability

The raw sequence data generated in this study have been deposited in the Genome Sequence Archive in National Genomics Data Center, Beijing Institute of Genomics (China National Center for Bioinformation), Chinese Academy of Sciences, under accession number CRA003454 that are publicly accessible at https://bigd.big.ac.cn/gsa (accessed on 23 April 2021).

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
