# Peer review of "Identification of Potential Key lncRNAs in the Context of Mouse Myeloid Differentiation by Systematic Transcriptomics Analysis"

_genes, 2021, doi:10.3390/genes12050630_

Round 1
Reviewer 1 Report
The authors Lan et al. have presented a study wherein they identified potential key lncRNAs in the context of myeloid differentiation by systematic transcriptomics analysis. Overall, they have done a very good job. Although I find there are some lacunae which should be improved.
Abstract:
The first line does not make any sense, and please modify it.
Please expand the abbreviation of 'lncRNAs' for the first time.
In addition, claims such as "great significance" are discouraged.
Introduction:
Please modify the introduction. Too much information is given, specifically in the second lncRNA paragraph. It is nice to be thorough, but it makes it too difficult for a reader to understand. So, please be concise and do not overwhelm the readers with information. Please consider bringing the third paragraph into the second position.
Materials and methods:
Several glaring mistakes are there in this section.
Why is the 'RNA-seq' section later? It should be the first one, right? Why did you use 'Poly-A RNAs'? While some of the lncRNAs appear indistinguishable from mRNAs, having 5' cap structures and 3' poly(A) tails, but most do not. So, it should have been the whole transcriptome with ribosomal depletion.
Why didn't you use the latest mouse genome (GRCm39)? Detection of lncRNAs is dependent on the genome assemblies used. Thus, always use the latest genomic assemblies and the well-curated annotation assemblies (Weirick et al., 2015).
Another major problem is the usage of outdated software. Why did you use Tophat2 (v2.0.13)? The latest release of TopHat was TopHat 2.1.1, released in 2016! Numerous publications (Ebrahim Sahraeian et al., 2017; Baruzzo et al., 2016, etc.) have categorically shown that TopHat is the worst one for aligning RNA-seq reads. Even the TopHat website page encourages scientists to use other tools (https://ccb.jhu.edu/software/tophat/index.shtml). Additionally, the Cufflinks toolset has also been deprecated and replaced by a more accurate StringTie.
All the bioinformatics work starting from the RNA-seq data analysis must be repeated with the current genome assembly and latest software.
Reviewer 2 Report
The manuscript by Lan and colleagues focusses on the expression and function of long non-coding RNAs during myeloid differentiation. Through generating expression data of diverse myeloid progenitors –inlcuding MLL-ENL expressing cells- and co-regulation analyses, the authors identified potential lncRNAs involved in murine myeloid differentiation. Following the identification, the authors focused on a new lncRNA, which they termed Gdal1 and could show inhibition of myeloid differentiation in a murine primary cell derived cell line model. Next, the authors aimed to unveil the transcriptional regulation of Gdal1 and identified 21 potential transcription factors with positive or negative correlation to Gdal1 expression. One candidate, Cepbe was analysed further and the authors provided convincing data for regulation of Gdal1 by CEBPe. Futrthermore, the authors aimed to identify potential functions of Gdal1 by gene expression analysis and provide data on CEBPa regulation after Gdal1 knoc-down.
The manuscript is overall sound and the authors provide convincing evidence for a functional role of Gdal1 in murine myeloid differentiation and its regulation. However, the functional data was exclusively obtained in a cell line model limiting its strength. Data on Cebpa regulations is very indirect and was obtained by comparison of more differentiated cells with less differentiated cells.
Overall, the manuscript will be of interest for the community and publication is highly recommended with some changes.
Major:
Please provide more direct data on Cebpa regulation by Gdal1. Currently, the data only shows that Cebpa is less upregulated in cells with blocked differentiation than in differentiating cells. This is to be expected but does not show that Gdal1 regulates Cebpa. Please show that knockdown in differentiated cells causes immediate reduction of Cebpa. Please adjust the heading and statements of paragraph 3.4 accordingly since it is currently overstating the findings.
Can the authors provide data in one additional cell model? While the study is sound, it may be limited by the cell model applied. Alternatively, use Hoxb8-immortalized cells or primary HSPCs to show differentiation block.
Did the authors check for conservation of Gdal1 (in particular in the human system)? Needs to be discussed.
The aspect of direct regulation needs to be discussed in more detail and critically.
The discussion initiates with a long repetition of the study. While it is common and useful to recapitulate the main findings, this paragraph needs to be shortened. Overall, the discussion is mainly repetition and should aim for more context in regard to current knowledge.
Figure 2i: How did the authors define the gating? Gr1+/CD11b- is an abnormal state and this reviewer wonders if the CSH3 cells are always a mixture of CD11b+ and CD11b- cells that upregulate Gr1 upon differentiation?
Minor:
The introduction may be shortened a bit on the side of transcription factor examples in differentiation.
Results section 1: It is unclear to this reviewer how the different myeloid cells were purified. Please provide more details on the procedure and markers for identifying the subpopulations
Figure 1c: Typo in the legend
Figure 4c: Typo in figure (Cabpa)
Round 2
Reviewer 1 Report
The authors have significantly improved the manuscript and I am happy with the current form. The MS should be accepted in its current form.
Reviewer 2 Report
I reviewed the revised version and the authors addressed all comments.
I fully support publication.